# Efficient Knocking Out of the Organophosphorus Insecticides Degradation Gene *opdB* in *Cupriavidus nantongensis* X1^T^ via CRISPR/*Cas9* with Red System

**DOI:** 10.3390/ijms24066003

**Published:** 2023-03-22

**Authors:** Yufei Zhang, Yuehan Geng, Shengyang Li, Taozhong Shi, Xin Ma, Rimao Hua, Liancheng Fang

**Affiliations:** 1Anhui Provincial Key Laboratory for Quality and Safety of Agri-Products, School of Resource & Environment, Anhui Agricultural University, Hefei 230036, China; 2Institute for Green Development, Anhui Agricultural University, Hefei 230036, China

**Keywords:** *Cupriavidus nantongensis* X1^T^, OPs degradation gene *opdB*, CRISPR/*Cas9*, Red system, gene editing

## Abstract

*Cupriavidus nantongensis* X1^T^ is a type strain of the genus *Cupriavidus*, that can degrade eight kinds of organophosphorus insecticides (OPs). Conventional genetic manipulations in *Cupriavidus* species are time-consuming, difficult, and hard to control. The clustered regularly interspaced short palindromic repeat (CRISPR)/associated protein 9 (*Cas9*) system has emerged as a powerful tool for genome editing applied in prokaryotes and eukaryotes due to its simplicity, efficiency, and accuracy. Here, we combined CRISPR/*Cas9* with the Red system to perform seamless genetic manipulation in the X1^T^ strain. Two plasmids, pACasN and pDCRH were constructed. The pACasN plasmid contained *Cas9* nuclease and Red recombinase, and the pDCRH plasmid contained the dual single-guide RNA (sgRNA) of organophosphorus hydrolase (*OpdB*) in the X1^T^ strain. For gene editing, two plasmids were transferred to the X1^T^ strain and a mutant strain in which genetic recombination had taken place, resulting in the targeted deletion of *opdB*. The incidence of homologous recombination was over 30%. Biodegradation experiments suggested that the *opdB* gene was responsible for the catabolism of organophosphorus insecticides. This study was the first to use the CRISPR/*Cas9* system for gene targeting in the genus *Cupriavidus*, and it furthered our understanding of the process of degradation of organophosphorus insecticides in the X1^T^ strain.

## 1. Introduction

The genus *Cupriavidus* is a class of Gram-negative bacteria that grows under chemoheterotrophic or chemolithotrophic conditions [1]. In recent years, several *Cupriavidus* species were isolated form soil, activated sludge, space equipment, root nodules of legumes, human clinical specimens and groundwater [2,3,4,5,6,7]. Heavy metal resistance is a representative characteristic of the genus *Cupriavidus* species [8]. Due to their significant prominence in pollutant bioremediation, biomineralization, symbiotic adaptation, biodetoxification, and the production of biodegradable polymers, *Cupriavidus* species have become a focus of scientific investigations [6,9,10,11,12]. For instance, *Cupriavidus* strains have been reported to degrade aromatic pollutants, including pesticides and halogenated phenols in various environments [13,14]. However, since there is not a model strain, efficient gene editing tools that are suitable for *Cupriavidus* species have rarely been reported, which has hindered the further exploration of the functions of these strains.

Traditional gene editing methods use a single recombinational insertion or allelic exchange of the flanking regions located upstream and downstream of the intergenic region to disrupt the open reading frames of the target gene in the genus *Cupriavidus* species [15,16,17,18,19]. However, complete biallelic knockouts can be achieved with relatively low mutation efficiency. The clustered regularly interspaced short palindromic repeat (CRISPR)/associated protein (Cas) system is an adaptive immune system in bacteria that works against mobile invading elements, such as phages and plasmids [20]. The CRISPR/*Cas9* system is a recently emerged genome editing tool that has high efficiency for genome modification, and it can be used to cause the complete abolishment of gene functions through the deletion of entire sequences or genes via homologous recombination (HR) [21]. Compared to ZFNs (zinc finger nucleases) and TALENs (transcription activator-like effector nucleases), the CRISPR/*Cas9* system is not time-consuming and labor-intensive to use [22]. CRISPR/*Cas9* was found in *Streptococcus pyogenes*, which could use a maturation CRISPR RNA (crRNA) and trans-activating crRNA (tracrRNA) to guide the protein of *Cas9* nuclease to the target region [23]. With the dissociation of mature crRNA and tracrRNA, a chimeric single-guide RNA (sgRNA) can be created and used favorably in genome editing [24]. Large indels can be produced by using dual sgRNAs that target one gene-guided *Cas9* nuclease to generate mutants in organisms, which would reduce the off-target rate and facilitate mutant screening through PCR [25,26,27]. In addition, *Cas9* endonuclease requires a specific protospacer-adjacent motif (PAM) of the “NGG” nucleotide sequence at the 3′ end of the sgRNA binding site in order to generate a double-strand break at the locus of interest in the target DNA [23,24].

The CRISPR/*Cas9* system has been used for genome editing in various bacterial species [28,29,30]. In the case of double strand breaks (DSBs) introduced by the CRISPR/*Cas9* complex, non-homologous end joining (NHEJ) and homologous recombination (HR) can be used to repair them to introduce indels. However, the NHEJ recombination machinery is either absent or does not function efficiently in most bacteria [31]. For the efficient and precise repair of DSBs, HR is an error-free system that uses an engineered homologous template in bacteria [32]. Furthermore, CRISPR/*Cas9* system combined with the λ-Red recombinant method can achieve multigene editing in *E. coli* simultaneously [33]. The efficiency of genome editing can be improved by using the λ-Red recombination system from phage recombinase-mediated homologous recombination [34]. The Red recombination system has a high homologous recombination efficiency because of its use of either single or double DNA fragments to introduce mutations into chromosomes, plasmids, and bacterial artificial chromosomes [35,36,37]. The Red system promotes the repair of double-strand breaks in DNA [38]. It consists of the Exo, Bet, and Gam proteins, which improve the frequency of insertions, deletions, and point mutations at the target locus specified by the flanking homology regions [35,37]. Exo is an exonuclease of the 5′-3′ end that cuts double-strand DNA (dsDNA). Bet stimulates the formation of joint molecules and strand exchange. When the system is stimulated by dsDNA ends, Gam can prevent DNA digestion of the λ phage and promote recombination so that Exo and Bet can approach the DNA ends [37].

In our previous study, a novel type of strain of the genus *Cupriavidus*, *C. nantongensis* X1^T^, was isolated and identified [39]. The X1^T^ strain could degrade eight kinds of organophosphorus insecticides (profenofos, chlorpyrifos, methyl parathion, parathion, triazophos, phoxim, fenitrothion and isocarbophos) [40]. Meanwhile, an organophosphorus degradation gene *opdB*, which encodes organophosphate hydrolase, was found through complete genome sequencing [39,41]. Here, we explored a stable and efficient method for gene editing in the X1^T^ strain by using CRISPR/*Cas9* with the Red system. Two plasmids were constructed for gene editing in the X1^T^ strain. The editing efficiency was evaluated by knocking out the organophosphorus insecticide degradation gene *opdB*. This may be a valuable genetic toolbox for the genus *Cupriavidus*.

## 2. Results

### 2.1. Construction and Screening of the CRISPR/Cas9-Based Genome Editing System in the X1^T^ Strain

To develop an efficient and convenient genetic manipulation tool for the X1^T^ strain, we introduced two plasmids into the X1^T^ strain for genome editing by harnessing the CRISPR/*Cas9*-based genome editing method. The RNA-guided nuclease *Cas9* from *Streptococcus pyogenes* formed the pACasN plasmid (Figure 1A). Then we combined dual spacers with homologous arms with a length of 600 bp in the pDCRH plasmid (Figure 1A) to assess the editing efficiency of the system in the X1^T^ strain. As is indicated in Figure 1B, there was a distinct difference in the number of transformants when another plasmid is introduced into the host cell. We calculated the transformation efficiency of two successive plasmids in the period of stable growth of the X1^T^ strain (Figure 1C). The transformation efficiency of pACasN was 86.1 ± 21.3 CFU/mg of donor. The transformation efficiency of pDCRH was 7.4 ± 1.9 CFU/mg of the donor. Another pDCRH plasmid assembled with dual sgRNAs and homologous repair arms was found to have a transformation efficiency that was 11.6-fold (86.1/7.4) lower than that when the pACasN plasmid bearing the *cas9* gene was transferred into the X1^T^ strain, which suggested that the plasmid-based expression of the *Cas9* protein encoded by the *cas9* gene had cleaved the target site in the plasmid of the X1^T^ strain efficiently.

### 2.2. Evaluation of the Efficiency of Gene Knockout of opdB in the X1^T^ Strain

To demonstrate the functionality of the CRISPR/*Cas9* system in the X1^T^ strain, we attempted to delete the whole ORF of *opdB*, which was 1.5 kb in length. As shown in Figure 2A, the sequence of the dual-sgRNA and homology repair arms was identified by bacteria solution PCR and Sanger sequencing. For the treatment of electroporation, the pACasN plasmid was transformed into the competent cells of the X1^T^ strain. Positive transformants were identified by colony PCR and Sanger sequencing (Figure 2B). The plasmid pDCRH-UD was transformed into the competent cells of the X1^T^ -pACasN strain. Positive transformants were identified by colony PCR and Sanger sequencing (Figure 2C). As indicated in Figure 2D, the PCR results indicated that the Δ*opdB* mutant strain was successfully obtained. The efficiency of knocking out *opdB* from the X1^T^ strain was over 30% (7/23). As shown in Figure 2E, the PCR product was obtained in the mutant strain after curing two plasmids, indicating the *opdB* gene was successfully deleted. As shown in Figure A1 in Appendix, Sanger sequencing was performed to confirm the deletion of the mutant strains of X1^T^-Δ*opdB*.

### 2.3. Plasmid Curing

For the curing of two plasmids, 6% *w*/*v* sucrose was used to induce the expression of the *sacB* gene. Then, cells were cultured at 30 °C for three days. The results showed that the induction of 6% *w*/*v* sucrose for three days caused much fewer colonies to grow on the plate than in the absence of sucrose (Figure 3A). The colonies of X1^T^-Δ*opdB* containing two plasmids were picked from the plate with 6% *w*/*v* sucrose and grown on the plates containing 100 μg/mL tetracycline (Figure 3B) and 50 μg/mL neomycin (Figure 3C), respectively. Curing is a consequence of protoplast divisions in which chromosomes are correctly partitioned, but plasmids are not. In the absence of the two plasmids harboring resistance markers, the X1^T^-Δ*opdB* strain was sensitive to two kinds of antibiotics. Therefore, several colonies could not grow on the plate with antibiotics. The experimental results indicated that both the pACasN and pDCRH plasmids were successfully cured. The curing efficiency of plasmids of pACasN and pDCRH was 13/16 and 4/16, respectively.

### 2.4. Assessment of the Degradation Capacity of OPs in the X1^T^-ΔopdB Strain

The degradation of 20 mg/L of eight kinds of OPs was assessed separately in the strains of X1^T^ and X1^T^-Δ*opdB*. It was found that eight kinds of OPs could be degraded and detected in 24 h by the X1^T^ strain as compared to the negative control (Table 1). On the other hand, the X1^T^-Δ*opdB* strain was unable to degrade these OPs as compared to the positive control of the X1^T^ strain, suggesting that the X1^T^-Δ*opdB* strain lost its functionality for degrading OPs. One-way ANOVA was used to compare the changes of eight kinds of organophosphorus insecticides residuals before and after the treatments of the strains of X1^T^ and X1^T^-Δ*opdB*. There was no significant difference when the significance level exceeded 5%. Different letters indicate significant differences between organophosphorus insecticides. The results demonstrate that the degradation of OPs was influenced by the *OpdB* gene in the X1^T^ strain.

## 3. Discussion

In this study, we report the development of the CRISPR/*Cas9*, an adaptive immune system from bacteria and archaea, has been widely used in genome editing in various species [28,29,42]. However, the expression of *Cas9* proteins is highly toxic, which may limit the possibility of easy transformation and the success of genome editing in various bacteria [43]. In our experiments, the X1^T^ strain with pACasN had stable characteristics that could be steadily inherited. This was enough to enable us to obtain desirable mutations and when two plasmids were transformed into the X1^T^ strain. Meanwhile, the efficiency of genome editing can be improved by using a λ-Red recombination system [34]. The genes associated with proteins of Red recombination and the *Cas9* nuclease expressed in the X1^T^ strain were first installed into a plasmid. From the experience of others to achieve a successful editing by expressing proteins before being prepared as the electrocompetent cells [33,44], we failed to isolate mutants of strain X1^T^. Maybe the proteins of Red recombination and *Cas9* nuclease are toxic that cannot exist in the X1^T^ strain for a long time. Therefore, we adopted the method of inducing proteins expression after the electroporation of two plasmids. Besides, the choices of highly efficient sgRNAs and the repair template length might have an impact on the efficiency of genome editing. Although there are many computational resources that can be used to design specific sgRNAs and predict off-target effects, their efficiency cannot be confirmed. When a single sgRNA was used, the genome editing efficiency was evaluated in the context of the same number of transformants, as well as dual sgRNAs [25,26,27]. We used the CRISPR/*Cas9* system in combination with a dual-sgRNA approach to target a gene simultaneously, which resulted in a loss of gene function and a minimization of possible off-target activity [26]. In addition, it was found there was no NHEJ repair mechanism in the X1^T^ strain via PCR screening and sequencing analysis. Besides, we should take the experimental selection of promoters for the expression of the *Cas9* nuclease and the sgRNA into consideration, so as to improve the deletion efficiency by more than 50% [45]. More detailed research will be carried out in further experiments.

It was previously reported that the X1^T^ strain is well known to have a great ability to degrade eight kinds of organophosphorus insecticides [40,41,46]. Organophosphate pesticides (OPs) can be degraded by various enzymes expressed in micro-organisms and mammals, including OP acid anhydrolase (OPAA) from rabbit tissue samples, OP-degrading hydrolases (OPHs) such as phosphotriesterase (PTE) and organophosphorus degradation enzyme (OPD), methyl parathion hydrolase (MPH) and glycerophosphodiesterase (GpdQ) from soil-dwelling microorganisms [47]. *OpdB* is a member of the novel subclass of OPD. The domain analyzed that *OpdB* has a hydrolase domain and its sequence identity with OPD was 52%. Both crude and purified *OpdB* enzymes were capable of degrading OPs [40,41]. The mechanisms of several OPs from eight kinds of organophosphorus insecticides were previously researched in the X1^T^ strain [40]. Due to the hydrolysis reaction of the initial step of the OPs in the X1^T^ strain, the *opdB* gene encoding OP hydrolase was found and identified through metabolite structure analysis of OPs [40]. Furthermore, the phenomenon of the knockout of the *opdB* gene in strain X1^T^ hindered the degradation of eight OPs, which further demonstrated that the *opdB* gene played an indispensable role. The results indicated that the CRISPR/*Cas9* is an attractive and powerful tool that can replace the classical gene knockout methods in *Cupriavidus* species. We believe that the two-plasmid system will be more widely adopted for further meaningful research.

## 4. Material and Methods

### 4.1. Strains and Plasmids

The strains and plasmids used in this study are listed in Table 2. The optimal growth conditions for the X1^T^ strain and all *E. coli* strains were Luria–Bertani medium at 37 °C. 

The complete genome sequences of the X1^T^ strain were previously sequenced and uploaded to GenBank database with the accession numbers of CP014844, CP014845 and CP014846.

### 4.2. Culture and Reagents 

Ampicillin (50 μg/mL), tetracycline (100 μg/mL) and neomycin (50 μg/mL) were added as needed. The MSM medium contained 1.0 g/L NaCl, 0.1 g/L MgSO_4_·7H_2_O, 0.3 g/L KH_2_PO_4_ and 1.32 g/L K_2_HPO_4_. Standard samples of profenofos (purity 98.0%), chlorpyrifos (purity 99.6%), methyl parathion (purity 99.5%), parathion (purity 99.5%), triazophos (purity 96.13%), phoxim (purity 99.5%), fenitrothion (purity 99.5%) and isocarbophos (purity 99.0%) were all purchased from Dr. Ehrenstorfer (Augsburg, Germany).

### 4.3. Construction of the Prokaryotic Expression Vector

The sequence of primers used in this study is listed in Table A1. The sequence and target sites of sgRNAs used in this study are listed in Table A2. The construction of the pACasN plasmid was performed by using the following method. The *cas9* gene was amplified from the p*Cas9* plasmid as previously reported [48]. The *araB* promoter (*P_araB_*) and Red recombination system were amplified from pKD20 [49]. All of the above DNA fragments were reconstructed into the *EcoR*I/*Hind*III sites in a pDN19 plasmid with a *T7* RNA polymerase promoter by using Gibson assembly [50]. The *sacB* gene, a widely used counter-selectable marker amplified from the pALB2 plasmid, was inserted into the *ApaL*I/*Xho*I sites of the preceding plasmid after editing [51]. The pACasN plasmid was constituted by these DNA fragments.

The pDCRH plasmid was constructed by using the following method. In the first step, a strong *E. coli* promoter (str) with the sgRNA and *sacB* gene was inserted into the *Kpn*I/*Hind*III sites of the broad-host-range plasmid vector pAK1900 [52]. Then, a neomycin resistance cassette from pAK405 was inserted into the *Sai*I site [53]. The resulting plasmid was named pDCRH. Two plasmids were then transformed into *E. coli* DH5α competent strains and stored at −80 °C.

The sgRNAs were designed by the CRISPOR design tool (http://crispor.tefor.net/ accessed on May 13th, 2022) without genomic alterations when analyzing the predicted off-targets in the X1^T^ strain [54]. The pDCRH plasmid that contained dual sgRNAs was constructed as described subsequently. Two oligonucleotides containing the dual-sgRNA arrays in the target gene of the X1^T^ strain were synthesized by Sangon Biotech (*Sangon*, Shanghai, China). Then, phosphorylation of oligos was carried out as follows: 5 μL 10 × T_4_ DNA ligase buffer (Takara, Kusatsu, Japan), 2 μL of dsgRNA-F, 2 μL of dsgRNA-R, 1 μL of T_4_ polynucleotide kinase (Takara), and 40 μL of ddH_2_O were added, followed by incubation at 37 °C for 1 h. Next, 2 μL of 1 M NaCl was added to the aforementioned mixed samples. Annealing procedures were performed as follows: 95 °C for 5 min; 95 °C for 30 s; 85 °C for 30 s, repeating 30 cycles and each cycle down by 2 degrees; 25 °C for 1 min, repeating 10 cycles and each cycle down by 0.1 degrees. The last solution was diluted 20-fold and stored at 4 °C. The pDCRH-spacer plasmid was constructed by inserting the dual sgRNA into the *Bsa*I site of pDCRH via Golden Gate assembly. The samples were mixed as follows: 1 μL of annealed oligos, 1 μL of pDCRH plasmid, 1 μL of T_4_ DNA ligase, 1 μL of 10 × T_4_ DNA ligase buffer, 1 μL of *Bsa*I-HF, and 5 μL of ddH_2_O. The reaction protocol was carried out in a thermal cycler at 37 °C for 3 min, 16 °C for 4 min, repeating 30 cycles, enzymes were inactivated at 65 °C for 15 min and slowly cooled down to 4 °C. The Golden Gate assembly product was transferred into 50 μL of *E. coli* DH5α competent cells. The cells were recovered in LB medium for 2 h at 37 °C and plated on an agar plate containing 50 μg/mL neomycin. The colonies were identified by PCR and sequencing.

The 600 bp DNA sequences of the upstream and downstream from the *opdB* gene were amplified from the X1^T^ strain by PCR. The PCR amplification was performed as follows: 95 °C for 3 min; 95 °C for 15 s, 58 °C for 15 s and 72 °C for 1.5 min, which was repeated for 34 cycles, followed by a final extension of 5 min at 72 °C and stored at 4 °C finally. The pDCRH-ΔUD plasmid was obtained by digesting the pDCRH-spacer plasmid with *Xba*I and *Xho*I endonucleases. The pDCRH-UD plasmid was constructed by inserting two homologous arms into the digested pDCRH-spacer plasmid through Gibson assembly. The steps of construction of pDCRH-UD followed the subsequent protocol. The reaction solution (1 μL upstream of the *opdB* gene, 1 μL downstream of the *opdB* gene, 5 μL of pDCRH-ΔUD plasmid, 3 μL of ddH_2_O, 10 μL of NEBuilder HiFi DNA Assembly Master Mix) was incubated at 50 °C for 30 min. The Gibson assembly product was transformed into 50 μL of *E. coli* DH5α competent cells. The cells were recovered in LB medium for 2 h at 37 °C and plated on an agar plate containing 50 μg/mL neomycin. The plasmid pDCRH assembled with the dual-sgRNA and homology arms was verified by PCR and sequencing.

### 4.4. Genome Editing

The competent cells of the X1^T^ strain were prepared as follows. The strain was activated and cultured in 20% concentration of LB medium at 37 °C for 12 h. Then, a total of 1 mL of cells was added to another fresh LB medium and incubated in a rotary sharker at 37 °C until the optical density at 600 nm reached 0.5. The cells were harvested by centrifugation at 5000 rpm for 15 min at 4 °C. Cell sediment was collected and washed six times with sterile ice-cold 10% *v*/*v* glycerol. The obtained cells were suspended with 0.5 mL sterile ice-cold 10% *v*/*v* glycerol in the end. The mechanism of genome editing with the two-plasmid system via CRISPR/*Cas9* is indicated in Figure 4. For electroporation, 90 μL of cells were mixed with 10 μL of pACasN series DNA at a concentration of 700 ng/μL and electroporation was performed in an ice-cold 2-mm Gene Pulser cuvette (Bio-Rad, Hercules, CA, USA) at 2.5 kV for 6 ms. Then, 500 μL of room-temperature LB medium was added and gently mixed. Cells were recovered at 30 °C for 3 h before being spread onto the LB plate (containing 100 mg/L tetracycline), and incubated at 30 °C for three days. Transformants were identified by PCR and sequencing. Next, a positive colony was picked from the plate and cultured in LB medium with a concentration of 20% at 30 °C until the optical density at 600 nm reached 0.5. The procedures for the preparation of X1^T^-pACasN competent cells were carried out according to the previous experimental methods. A total of 10 μL of pDCRH series DNA at a concentration of 700 ng/μL and 90 μL of X1^T^-pACasN competent cells were co-electroporated under the same circumstances. Cells were recovered at 30 °C for 3 h before being spread on the LB plate containing 100 mg/L tetracycline and 50 mg/L neomycin. Then, cells were incubated at 30 °C for eight days. Meanwhile, we attempted to perform Red induction by streaking the identified bacteria on LB plates containing L-arabinose (final concentration of 20 mM). Transformants were incubated at 30 °C for three days. The editing efficiency was calculated based on the number of mutant colonies divided by the number of colonies evaluated in the end.

### 4.5. Plasmid Curing

To cure the two-plasmid system after editing, the effects of sucrose concentration and incubation time were measured. Sucrose at a concentration of 6% *w*/*v* was chosen, and the incubation time was three days until the bacteria growth was evident. The identified bacteria were streaked onto LB plates containing tetracycline and neomycin and were cultured at 30 °C for 48 h. One colony from the X1^T^-Δ*opdB* strain containing pDCRH and pACasN was cultured in 2 mL of LB medium at 30 °C for 48 h. Then, 10 μL of cells were diluted 100× in fresh LB medium, and 100 μL of diluted culture was plated on the LB agar in the presence or absence of 6% *w*/*v* sucrose at 30 °C for 72 h. Sixteen colonies were randomly picked and streaked on three different LB plates with no antibiotics (culture collection, 100 μg/mL tetracycline, and 50 μg/mL neomycin). The efficiency of plasmid curing could be calculated by dividing the number of streaked colonies by the total number of colonies. The elimination of two plasmids was verified by bacteria solution PCR.

### 4.6. Degradation Experiments

The wild-type X1^T^ strain was incubated in 100 mL LB medium at 37 °C and the mutant colony was cultured in the same conditions. Each overnight culture was diluted by 1:100 in 100 mL of adjustive LB medium and incubated at 37 °C for 8 h. The strains were harvested by centrifugation at 7000 rpm for 10 min and washed three times with MSM medium. Cultures were diluted to an OD_600_ of 0.6. Degradation experiments of 8 kinds of organophosphorus insecticides were performed in 25 mL glass tubes with secure tops. A total of 20% of the cells in 5 mL of culture were incubated in MSM at 37 °C. The samples were regularly sampled by adding 5 mL of acetonitrile to terminate the reaction and extract pesticides. The residuals of the OPs were measured via HPLC (Agilent 1260, Waldbronn, Germany) method previously reported [40].

## 5. Conclusions

This study first reported that the CRISPR/*Cas9* coupled with Red homologous recombination can be used to perform genome editing in the genus *Cupriavidus*. The two-plasmid system is an efficient and accurate method to realize the specific manipulation of the genome in *C. nantongensis* X1^T^. The results showed that the editing efficiency was over 30%. The biodegradation experiments on OPs revealed that the *opdB* gene was responsible for the catabolism of OPs in the X1^T^ strain. This method of using CRISPR/*Cas9* in combination with the Red recombination system has laid the foundation for further research on *Cupriavidus nantongensis* X1^T^.

## Figures and Tables

**Figure 1 ijms-24-06003-f001:**
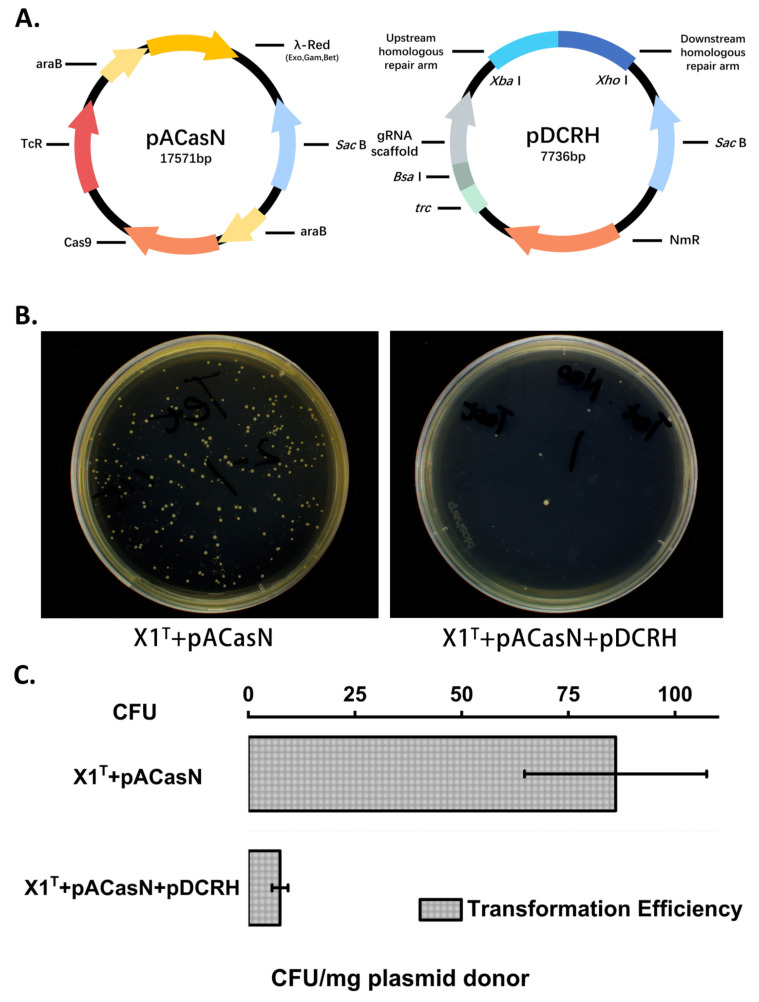
The transformation of two plasmids into strain X1^T^ was performed by using electroporation. (**A**) Maps of two plasmids of pACasN and pDCRH. *araB*, an inducible promoter driven by the L-arabinose; *SacB*, a counter-selectable marker used for plasmid curing after editing; *Tet*, a tetracycline-resistance marker in the X1^T^ strain; *trc*, a strong promoter to express the dual sgRNA; *Bsa*I, used as the spacers via Golden Gate assembly; *Xba*I and *Xho*I, used as the homologous repair arms via Gibson assembly; *Neo*, a neomycin-resistance marker in the X1^T^ strain. (**B**) The plasmid pACasN (**Left**) was electroporated into the strain X1^T^. The plasmid pDCRH (**Right**) was electroporated into the strain X1^T^-pACasN. (**C**) The transformation efficiency of two plasmids was assessed by the number of colonies of strain X1^T^.

**Figure 2 ijms-24-06003-f002:**
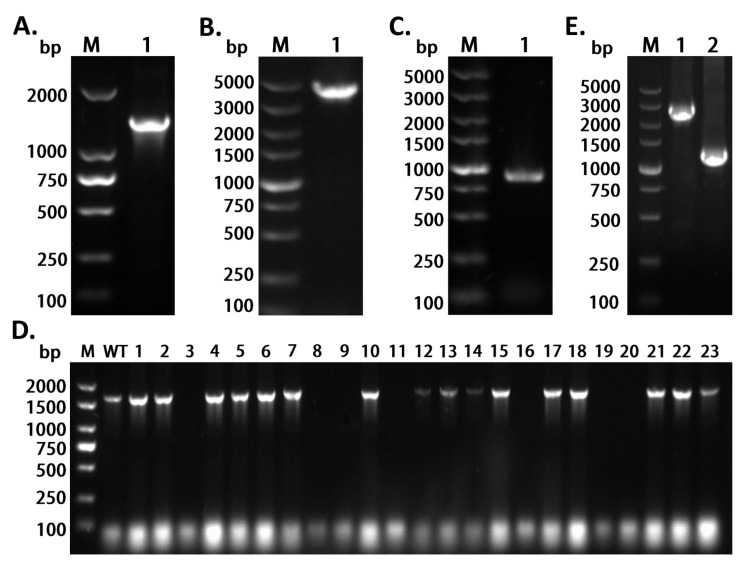
The two-plasmid system enabled the deletion of *opdB* in the X1^T^ strain. (**A**) Accurate repair of DSBs was performed by binding the entire homology repair arms and the dual sgRNA with primers P1/P2 in the pDCRH plasmid. The length was 1.4 kb. (**B**) Following electroporation of the pACasN plasmid, the 4.1-kb DNA fragment of the *cas9* gene was identified by the primer pair *Cas9*-F/R in the X1^T^ strain. (**C**) A marker gene of *Neo* (neomycin resistance gene) of pDCRH was amplified by colony PCR using the primer pair Neo-F/R in the X1^T^ strain harboring two plasmids. (**D**) CRISPR/*Cas9*-based disruption of the *opdB* gene in the X1^T^ strain with primers *opdB*-F/R. The WT is a positive control. The efficiency of knocking out *opdB* from *Cupriavidus nantongensis* X1^T^ was 7/23. (**E**) Verification of the *opdB* gene knockout with primers T600-F/B600-R in the strains of X1^T^ and X1^T^-Δ*opdB* after curing two plasmids. The length of the deleted gene of *opdB* was 1.5 kb.

**Figure 3 ijms-24-06003-f003:**
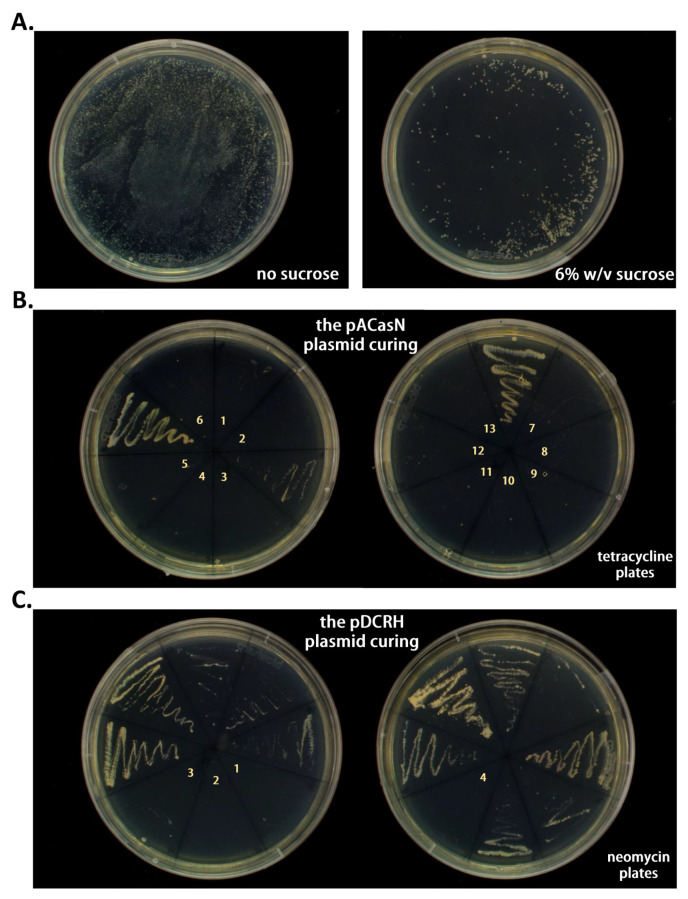
The pACasN/pDCRH system can be easily cured after editing. (**A**) Fewer colonies were survived in 6% *w*/*v* sucrose at 30 °C for 72 h in the presence of 6% *w*/*v* sucrose (**right**) than that in the absence of sucrose (**left**). (**B**) The plasmid pACasN was cured with an efficiency of 13/16. The yellow numbers represent the number of colonies that did not grow on plates with tetracycline. (**C**) The plasmid pDCRH was cured with an efficiency of 4/16. The yellow numbers represent the number of colonies that did not grow on plates with neomycin.

**Figure 4 ijms-24-06003-f004:**
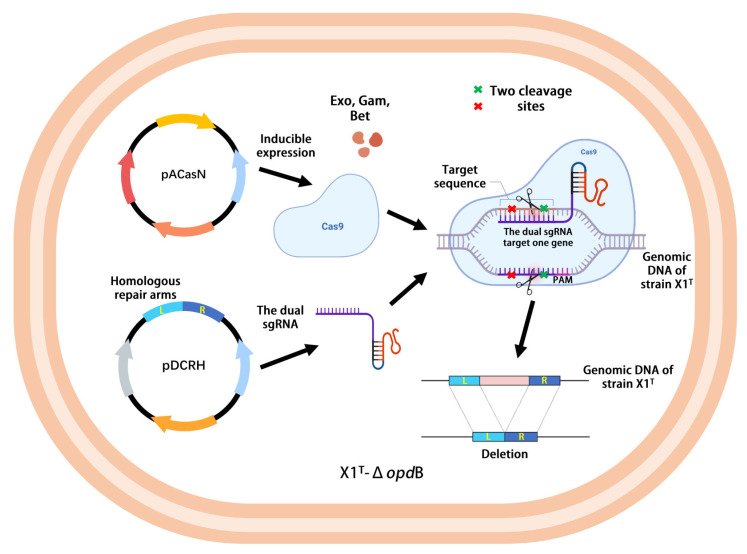
The mechanism of genome editing with the two-plasmid system via CRISPR/*Cas9*.

**Table 1 ijms-24-06003-t001:** Degradation of eight kinds of OPs at the concentration of 20 mg/L by the strains of X1^T^ and X1^T^-Δ*opdB*.

OPs	Concentration of OPs after 24 h of Treatment (mg/L)		
	Control	*C. nantongensis* X1^T^	*C. nantongensis* X1^T^-Δ*opdB*
Profenofos	17.1 ± 0.4 ^c^	9.7 ± 0.2 ^a^	17.7 ± 0.3 ^c^
Chlorpyrifos	19.1 ± 0.1 ^ab^	ND	19.4 ± 0.4 ^ab^
Methyl parathion	19.6 ± 0.0 ^a^	ND	19.7 ± 0.4 ^a^
Parathion	18.7 ± 0.5 ^b^	ND	18.7 ± 0.1 ^b^
Triazophos	18.6 ± 0.2 ^b^	1.4 ± 0.1 ^c^	18.6 ± 0.2 ^b^
Phoxim	18.5 ± 0.4 ^b^	1.0 ± 0.0 ^d^	18.7 ± 0.3 ^b^
Fenitrothion	17.6 ± 0.4 ^c^	ND	17.0 ± 0.4 ^c^
Isocarbophos	14.2 ± 0.3 ^d^	6.2 ± 0.1 ^b^	14.0 ± 0.2 ^d^

ND: cannot detected, meaning that the value was below the detection limit. Different letters: meaning significant differences among eight kinds of OPs (*p* ≤ 0.05).

**Table 2 ijms-24-06003-t002:** The bacterial strains and plasmids used in this study.

Strains and Plasmids	Description	Source
*E. coli DH5α*	supE44ΔlacU169 (φ80 lacZΔM15) hsdR17 recA1 end A1 gyrA96 thi-1 relA1	Solarbio
pMD19-T	Ap^r^	Takara
*Cupriavidus nantongensis* X1^T^	an aerobic, Gram-negative, motile Proteobacterium that forms circular colonies	Lab store
p*Cas9*	Cm^r^, bacterial expression of *Cas9* nuclease; a temperature sensitive vector in *E. coli* for genome editing	BioSci
pKD20	the Red recombination system and the arabinose-inducible P_araB_ promoter	BioSci
pDN19	Tc^r^, the *P. aeruginosa* shuttle cloning vector	BioSci
pALB2	Tc^r^, SacB	BioSci
pAK1900	Ap^r^, broad-host-range cloning vector	BioSci
pAK405	Plasmid for allelic exchange and seamless gene deletions; Nm^r^	BioSci
pACasN	A shuttle vector; Tc^r^, SacB, *Cas9* nuclease and the Red recombination system	This study
pDCRH	A shuttle vector; Nm^r^; SacB	This study
pDCRH-spacer	The pDCRH plasmid with the dual sgRNA	This study
pDCRH-ΔUD	Digested pDCRH-spacer plasmid	This study
pDCRH-UD	The pDCRH plasmid with the dual sgRNA and homologous arms	This study
X1-pACasN	Tc^r^	This study
X1-Δ*opdB*	The *opdB* gene deleted in *Cupriavidus nantongensis* X1^T^	This study

## Data Availability

Not applicable.

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
