# Peer review of "Efficient Knocking Out of the Organophosphorus Insecticides Degradation Gene opdB in Cupriavidus nantongensis X1T via CRISPR/Cas9 with Red System"

_ijms, 2023, doi:10.3390/ijms24066003_

Round 1

Reviewer 1 Report

The CRISPR/Cas9 system was used in this study to target specific genes in the Cupriavidus genus species. Additionally, the manner in which strain X1T's organophosphorus insecticides were degraded was addressed. In spite of the significant value of this work, some clarification is urgently needed. here are some concerns:

1. The title is too long and does not convey the scientific soundness of the study in a clear and concise manner.

2.  Abstract: Many acronyms, such as CRISPR/Cas9, pDCRH, sgRNA, etc., were used without their full definitions being stated. The full term should be clarified on their first mention

3. What is Red system? please clarify and describe properly

4. Line 67:  ""It consists of proteins of Exo, Bet, and Gam, . please clarify and describe

5. In line 75, the authors discuss a stable and efficient gene editing strategy using CRISPR/Cas9 and Red systems in strain X1T. What is the strategy the authors used? Generally, study objectives are not clearly presented.

6. A lot of work needs to be done on the language in general. I found many typographical errors overall the manuscript.

Author Response

The CRISPR/Cas9 system was used in this study to target specific genes in the Cupriavidus genus species. Additionally, the manner in which strain X1T's organophosphorus insecticides were degraded was addressed. In spite of the significant value of this work, some clarification is urgently needed. here are some concerns:

  1. The title is too long and does not convey the scientific soundness of the study in a clear and concise manner.

Response: Thanks for the title suggested. Your comments are really thoughtful. We have changed the title of “Stable and efficient gene editing strategy for the knockout of the organophosphorus insecticides degradation gene opdB in Cupriavidus nantongensis X1T via CRISPR/Cas9 with Red system” into “Efficient genetic editing of knocking out of organophosphorus insecticides degradation gene opdB in Cupriavidus nantongensis X1T via CRISPR/Cas9”. Thank you for your comments.

  1. Abstract: Many acronyms, such as CRISPR/Cas9, pDCRH, sgRNA, etc., were used without their full definitions being stated. The full term should be clarified on their first mention

Response: Thank you for your professional reviews. We have clarified the “CRISPR/Cas9” at line 11 of Page 1, the“pACasN and pDCRH”at Line 14 of Page 1 on their first mention. The “sgRNA” has been clarified at Line 16 of Page 1 on the first mention. Thank you for your comments.

  1. What is Red system? please clarify and describe properly

Response: Thank you for your professional reviews. The Red recombination system comes from phage. It has high efficiency homologous recombination by using either single or double DNA fragments to introduce mutations into chromosomes, plasmids and bacterial artificial chromosomes. The Red system can promote the repair of DNA double-strand breaks. The Red system consists of proteins of Exo, Bet, and Gam, which improves the creation frequency of insertions, deletions, and point mutations at the target locus specified by flanking homology regions. Exo is an exonuclease of 5’-3’ end that cuts double-strand DNA (dsDNA). Bet stimulates the formation of joint molecules and strand exchange. When the system is stimulated by dsDNA ends, Gam can prevent DNA digestion of λ phage and promote recombination so that Exo and Bet can approach DNA ends. We have futher clarified at Line 94-98 of Page 2. Thank you for your comments.

  1. Line 67:  ""It consists of proteins of Exo, Bet, and Gam, please clarify and describe.

Response: Thank you for your professional reviews. We have further clarified at Line 133-137 of Page 2. Exo is a 5’-3’ exonuclease that acts on double-stranded (ds) DNA processively. Bet is a single-stranded DNA (ssDNA) binding protein capable of annealing complementary ssDNA strands. Bet can stimulate the formation of joint molecules and strand exchange. Gam’s role is to prevent RecBCD-promoted digestion of phage DNA, so that Exo and Bet can gain access to DNA ends to promote recombination [1]. Thank you for your comments.

[1] Murphy, K. C., Use of Bacteriophage λ Recombination Functions To Promote Gene Replacement in Escherichia coli. J Bacteriol. 1998, 180, 2063-2071.

  1. In line 75, the authors discuss a stable and efficient gene editing strategy using CRISPR/Cas9 and Red systems in strain X1T. What is the strategy the authors used? Generally, study objectives are not clearly presented.

Response: Thank you for your professional comments. We sincerely thank the reviewer with your suggestions and have further clarified and made a few changes to better express our study objectives. We described “a stable and efficient gene editing strategy using CRISPR/Cas9 and Red systems in strain X1T” into “Here, we explored a stable and efficient method for gene editing in the X1T strain by using CRISPR/Cas9 with the Red system. Two plasmids were constructed for gene editing in the X1T strain. The editing efficiency was evaluated by knocking out the organophosphorus insecticide degradation gene opdB. This may be a valuable genetic toolbox for the genus Cupriavidus.” at Line 144-148 of Page 3. Thank you for your comments.

  1. A lot of work needs to be done on the language in general. I found many typographical errors overall the manuscript.

Response: Thank you for your professional review work on our article.  We sincerely thank the reviewer for careful reading. Extensive corrections of language and typographical errors have been made to previous draft. In our resubmitted manuscript, the mistakes are revised in the red text. Thank you for your comments.

Reviewer 2 Report

In the Cupriavidus research field, it should be an interesting paper. I am just wondering whether you could add to knock out one more gene except opdB to show that the systems you developed work well? The English writing needs approve before acceptance.

Author Response

In the Cupriavidus research field, it should be an interesting paper. I am just wondering whether you could add to knock out one more gene except opdB to show that the systems you developed work well? The English writing needs approve before acceptance.

Response: Thank you for your professional comments. We agree with you that more study to knock out one more gene to assess the widespread use of our genomic editing system. Actually, our recent work has been made to find more interesting phenomena. However, molecular biology experiment cycle is long. More details cannot be presented immediately.

We sincerely thank you for your valuable feedback that we have used to improve our research.

In addition, we regret there were problems with the English. The paper has been carefully revised in our resubmitted manuscript. Thank you very much for your comments and suggestions.

Reviewer 3 Report

There are very interesting approach and relatively well writen story.

 However, some corrections are required before acceptance.

Deatils:

Line 18, 19: why was? Should be present tense.

Line 20: Further the understanding?

Line 28: from or form?

Line 33: “an excellent focus” ?

Line 137: “Several colonies cannot grow on the plate with antibiotics” – please, clarify.

Author Response

There are very interesting approach and relatively well writen story.

 However, some corrections are required before acceptance.

Resopnse: We feel great thanks for your professional review work on our article. As you are concerned, there are several problems that need to be addressed. According to your nice suggestions, we have made extensive corrections to previous draft. In our resubmitted manuscript, the mistakes are revised.

Deatils:

Line 18, 19: why was? Should be present tense.

Resopnse: Thanks for your reminder. We are really sorry for our careless mistakes. We have revised the tense “was” into “is” (Page 1, Line 18 and 19)

Line 20: Further the understanding?

Resopnse: Thanks for your reminder. We are really sorry for our careless mistakes. We have corrected “the” into “our”. (Page 1, Line 21)

Line 28: from or form?

Resopnse: Thanks for your careful checks. We are really sorry for our careless mistakes. We have corrected the “form” into “from”. (Page 1, Line 29)

Line 33: “an excellent focus” ?

Resopnse: Thanks for your reminder. We are really sorry for our careless mistakes. We have deleted the “an”. ( Page 1, Line 82). Thanks for your professional review work.

Line 137: “Several colonies cannot grow on the plate with antibiotics” – please, clarify.

Resopnse: Thanks for your professional review work. We have carefully considered the question of line 137. To our knowledge, the use of sacB as a counter-selectable marker confers sucrose sensitivity upon gram-negative bacteria. The activity of levansucrase, an enzyme encoded by the Bacillus subtilis sacB gene, which is inducible in the presence of sucrose, results in the synthesis of lethal amounts of levan and the accumulation of levan in the periplasm, causing cell lysis [1]. However, allelic-exchange mutants to sucrose tolerance occurred at a low frequency, making some potential counter-selectable donor strains for gene transfer studies [2]. Protoplast divisions are necessary for plasmid curing. Curing is a consequence of divisions in which chromosomes are correctly partitioned but plasmids are not [3]. In the absence of the two plasmids harboring resistance markers, strain X1TopdB is sensitive to two kinds of antibiotics. Therefore, several colonies cannot grow on the plate with antibiotics. We have made some changes in the manuscript of Line 272 of Page. Thank you for your suggestions and comments. (Page 6, Line 272)

[1] GAY, P.; CQQ, D. L.; STEINMETZ, M.; BERKELMAN, T.; KADOl, C. I., Positive Selection Procedure for Entrapment of Insertion Sequence Elements in Gram-Negative Bacteria. J Bacteriol. 1985, 164, 918-921.

[2] Blomfield, I. C.; Vaughn, V.; Rest, R. F.; Eisenstein, B. I., Allelic exchange in Escherichia coli using the Bacillus subtilis sacB gene and a temperature-sensitive pSC101 replicon. Mol Microbiol. 1991, 5, 1447-57.

[3] Gruss, A.; Novick, R., Plasmid Instability in Regenerating Protoplasts of Staphylococcus aureus Is Caused by Aberrant Cell Division. J Bacteriol. 1986, 165, 878-883.

Reviewer 4 Report

The manuscript demonstrates successful seamless genetic manipulation of Cupriavidus nantongensis X1T, using the CRISPR/Cas9 system and the Red system. The authors constructed two plasmids to delete the target gene opdB, which is responsible for organophosphorus insecticide catabolism. The study is the first to utilize the CRISPR/Cas9 system for gene targeting in the genus Cupriavidus, and it provides insight into the degradation process of organophosphorus insecticides in strain X1T.

However, there are several issues that need to be addressed:

1. The authors need to perform statistical analyses for their presented data.

2. The Cas9-mediated gene editing efficiency needs to be studied by sequencing.

3. The toxicity of Cas9 needs to be evaluated.

Author Response

The manuscript demonstrates successful seamless genetic manipulation of Cupriavidus nantongensis X1T, using the CRISPR/Cas9 system and the Red system. The authors constructed two plasmids to delete the target gene opdB, which is responsible for organophosphorus insecticide catabolism. The study is the first to utilize the CRISPR/Cas9 system for gene targeting in the genus Cupriavidus, and it provides insight into the degradation process of organophosphorus insecticides in strain X1T.

However, there are several issues that need to be addressed:

  1. The authors need to perform statistical analyses for their presented data.

Response: Thank you for your professional review work on our article. One-way ANOVA was adopted to compare the changes of 8 kinds of organophosphorus insecticides residuals before and after the treatments of strain X1T and X1TopdB. There was no significant difference when the significance level exceeded 5%. Different letters indicate significant differences between organophosphorus insecticides. According to your nice suggestions, we have made further clarified in our resubmitted manuscript at Line 290-300 of Page 7. Thank you for your comments.

  1. The Cas9-mediated gene editing efficiency needs to be studied by sequencing.

Response: Thank you for your professional review work on our article. The deletion of Mutant strains of X1TopdB was confirmed by sanger sequencing in Supplement Figure A1 at the end of our manuscript. We have made some changes in our languages at Line 625 of Page 13. Thank you for your comments.

  1. The toxicity of Cas9 needs to be evaluated.

Response: Thank you for your professional review work on our article. As far as you are concerned, Cas9 can be inherently toxic, which has limited its use in some organisms. Cas9-mediated toxicity has also been described in other organisms. For example, Benedikt M. Markus et al sought to define the requirements for stable Cas9 expression, comparing different expression constructs and characterizing the role of the buffering sgRNA to understand the basis of Cas9 toxicity [1]. Their work also emphasizes the need for further characterizing the effects of Cas9 expression. As with other in vivo methods, CRISPR/Cas9 cannot circumvent the host toxicity issue. Cas9 toxicity has been observed in many prokaryotes as well as other eukaryotes. Modification of Cas9 by shortening its half-life can help develop CRISPR/Cas9-based genome editing approaches, maintain the genome editing capacity and alleviate cell toxicity [2]. Previous studies have generated modified versions of Cas9 with fast turnover in vitro or chemically controlled split-Cas9 [3-5]. It is imperative to clarify the influence of Cas9 on strain X1T and to alleviate its potentially detrimental effects. The correlation between Cas9 expression and C. nantongensis X1T will be determined in our subsequent investigations. Therefore, the referee’s concern is of importance for our further study. Thank you for your comments.

[1] Markus, B. M.; Bell, G. W.; Lorenzi, H. A.; Lourido, S., Optimizing Systems for Cas9 Expression in Toxoplasma gondii. msphere. 2019, 4, e00386-19.

[2] Yang, S.; Li, S.; Li, X. J., Shortening the Half-Life of Cas9 Maintains Its Gene Editing Ability and Reduces Neuronal Toxicity. Cell Rep. 2018, 25, 2653-2659.

[3] Gutschner, T.; Haemmerle, M.; Genovese, G.; Draetta, G. F.; Chin, L., Post-translational Regulation of Cas9 during G1 Enhances Homology-Directed Repair. Cell Rep. 2016, 14, 1555-1566.

[4] Tu, Z.; Yang, W.; Yan, S.; Yin, A.; Gao, J.; Liu, X.; Zheng, Y.; Zheng, J.; Li, Z.; Yang, S.; Li, S.; Guo, X.; Li, X. J., Promoting Cas9 degradation reduces mosaic mutations in non-human primate embryos. Sci Rep. 2017, 7, 42081.

[5] Zetsche, B.; Volz, S. E.; Zhang, F., A split-Cas9 architecture for inducible genome editing and transcription modulation. Nat Biotechnol. 2015, 33, 139-142.

Round 2

Reviewer 3 Report

Minor corrections are required. some of them are below:

Lines 227-229: please, edit sentence grammar.

Lines 320- 323: you do not need to mention in each acse starin name, you use only one.

Line 370: „followed by immediate suspension“ – this is not correct. Do you mean mixing?

Author Response

Minor corrections are required. some of them are below:

  1. Lines 227-229: please, edit sentence grammar.

Response: As recommended, we have changed the sentence to “It was previously reported that the X1T strain is well known to have a great ability to degrade eight kinds of organophosphorus insecticides.”. (Line 227, Page 8)

  1. Lines 320- 323: you do not need to mention in each acse starin name, you use only one.

Response: Yes. We have revised the sentence to “The competent cells of the X1T strain were prepared as follows. The strain was activated and cultured in 20% concentration of LB medium at 37 °C for 12 h. Then, a total of 1 mL of cells were added to another fresh LB medium and incubated in a rotary sharker at 37 °C until the optical density at 600 nm reached 0.5.”. (Line 319, Page 10)

  1. Line 370 : „followed by immediate suspension“ – this is not correct. Do you mean mixing?

Response: Thank you for your careful checks. As recommended, we have revised this sentence to “For electroporation, 90 μL of cells were mixed with 10 μL of pACasN series DNA at a concentration of 700 ng/μL and electroporation was performed in an ice-cold 2-mm Gene Pulser cuvette (Bio-Rad) at 2.5 kV for 6 ms. Then, 500 μL of room-temperature LB medium was added and gently mixed.”. (Line 330, Page 11)
